# Evaluation of Static and Dynamic Residual Mechanical Properties of Heat-Damaged Concrete for Nuclear Reactor Auxiliary Buildings in Korea Using Elastic Wave Velocity Measurements

**DOI:** 10.3390/ma12172695

**Published:** 2019-08-23

**Authors:** Seong-Hoon Kee, Jun Won Kang, Byong-Jeong Choi, Juho Kwon, Ma. Doreen Candelaria

**Affiliations:** 1Department of Architectural Engineering, Dong-A University, Busan 49315, Korea; 2Department of Civil Engineering, Hongik University, Seoul 04066, Korea; 3Department of Architectural Engineering, Kyonggi University, Suwon 16227, Korea; 4Department of Civil Engineering, Korea Electric Power Corporation Engineering and Construction (KEPCO E&C), Gimcheon-si 39660, Korea

**Keywords:** concrete, thermal-induced damage, mechanical properties, non-destructive evaluation

## Abstract

The main objectives of this study are (1) to investigate the effects of heating and cooling on the static and dynamic residual properties of 35 MPa (5000 psi) concrete used in the design and construction of nuclear reactor auxiliary buildings in Korea; and (2) to establish the correlation between static and dynamic properties of heat-damaged concrete. For these purposes, concrete specimens (100 mm × 200 mm cylinder) were fabricated in a batch plant at a nuclear power plant (NPP) construction site in Korea. To induce thermal damages, the concrete specimens were heated to target temperatures from 100 °C to 1000 °C with intervals of 100 °C, at a heating rate of 5 °C/min and allowed to reach room temperature by natural cooling. The dynamic properties (dynamic elastic modulus and dynamic Poisson’s ratio) of concrete were evaluated using elastic wave measurements (P-wave velocity measurements according to ASTM C597/C597M-16 and fundamental longitudinal and transverse resonance tests according to ASTM C215-14) before and after the thermal damages. The static properties (compressive strength, static elastic modulus and static Poisson’s ratio) of heat-damaged concrete were measured by the uniaxial compressive testing in accordance with ASTM C39-14 and ASTM C469-14. It was demonstrated that the elastic wave velocities of heat-damaged concrete were proportional to the square root of the reduced dynamic elastic moduli. Furthermore, the relationship between static and dynamic elastic moduli of heat-damaged concrete was established in this study. The results of this study could improve the understanding of the static and dynamic residual mechanical properties of Korea NPP concrete under heating and cooling.

## 1. Introduction

Auxiliary buildings of a nuclear reactor are often located adjacent to the reactor containment structure in a nuclear power plant (NPP). These buildings house most of the auxiliary and safety systems associated with the reactor, such as radioactive waste systems, chemical and volume control systems, and emergency cooling water systems. Many parts of the structures, systems and components (SSC) in NPP auxiliary buildings are made of concrete materials that help in disaster prevention by minimizing environmental impacts. This is achieved by preventing outflow of radioactive materials in extreme conditions such as nuclear accidents. In addition, concrete in the structures provides the environment necessary for operation of NPPs by protecting SSC in service state. Unlike typical structures, concrete in NPPs could be exposed to heating and cooling conditions in a variety of temperature environments. From the perspective of safe and reliable operation of NPPs, it is important to understand the variation of mechanical properties of NPP concrete according to heating and cooling actions, to develop an effective method for the condition assessment of heat-damaged concrete in structures, and, if needed, to decide proper maintenance strategies [1].

There are various testing methods for evaluating heat-induced damages in concrete. An initial level of the evaluation relies primarily on visual inspection and sounding (or hammer tapping). However, visual inspections provide only superficial information and the results depend on the experience of the inspector. The sounding method is subjective in nature and requires experienced inspectors to obtain accurate results. Furthermore, assessments of local components based on destructive tests (or invasive test) have been used to acquire more detailed information on physical and chemical changes in heat-damaged concrete. Various analysis techniques could be used in the laboratory (e.g., differential thermal analysis, dilatometry, porosimeter, colorimetry, micro-crack density analysis, X ray diffraction analysis, chemical analysis) [2]. However, they need point-by-point response of small samples extracted from concrete structures. Thus, they are labor-intensive and time-consuming, and cannot be applied ubiquitously over the entire area of the structure. Many researchers have developed various non-destructive test (NDT) methods, which can be applied rapidly to heat-damaged concrete in structures. Previous researches have summarized the advantages and limitations of NDT methods normally applied to the condition assessment of heat-damaged concrete [2].

The NDT methods based on elastic wave velocity measurements (resonance vibration test, ultrasonic pulse velocity test, impact-echo, surface wave measurements, etc.) have gained popularity for the condition assessment of heat-damaged concrete, both in the field and laboratory. Some researchers have evaluated the dynamic elastic modulus of heat-damaged concrete by measuring the resonance frequency of cylindrical or prismatic concrete in the laboratory [3,4,5,6]. Some researchers have demonstrated that P-wave velocity of concrete is sensitive to physical and chemical changes in concrete after being exposed to high temperatures (e.g., dehydration of C-S-H gel, decomposition of calcium hydroxide, and micro-cracking developed in interfacial transition zone of concrete) [7,8,9]. In addition, some researchers have also measured the resonance frequency of circular, thin-disk shaped concrete obtained from core samples using the impact-echo test for post-fire condition assessment of concrete structures [10,11,12]. 

It is of importance to obtain the correlation between static and dynamic mechanical properties of heat-damaged concrete for the reliable condition assessment of concrete in structures exposed to various heating and cooling conditions. Static mechanical properties (static elastic modulus, compressive strength and static Poisson’s ratio) of heat-damaged concrete have been studied extensively in the literature and summarized by previous researchers [1,13,14]. However, there is a gap of knowledge on the dynamic mechanical properties (e.g., dynamic elastic modulus and dynamic Poisson’s ratio) of heat-damaged concrete based on elastic wave velocity measurements due to a lack of experimental data. The elastic wave velocity tests are based on measuring the velocity of elastic stress waves (compressive or transverse waves). The mechanical properties evaluated, based on elastic wave velocity measurements, are known as dynamic mechanical properties (dynamic elastic modulus and dynamic Poisson’s ratio). The compressive wave velocities in unbounded solid media ($V_{p}$) and thin rods ($V_{c}$) are called constrained and unconstrained compressive wave velocities, respectively. For elastic homogeneous and isotropic media, the two compressive wave velocities in the two different confinement conditions are expressed as the following equations [15]: (1)$$V_{p}=\sqrt{\frac{E_{d}\left(1-\upsilon_{d}\right)}{\rho \left(1+\upsilon_{d}\right)\left(1-2\upsilon_{d}\right)}},$$
(2)$$V_{c}=\sqrt{\frac{E_{d}}{\rho }},$$
where $E_{d}\;$ is dynamic elastic modulus of concrete; $\upsilon_{d}$ is dynamic Poisson’s ratio and ρ is mass density of concrete. In contrast, transverse wave velocity (also known as S-wave velocity) in the test media can be expressed, without regard to the confinement conditions of media, as follows,
(3)$$V_{s}=\sqrt{\frac{E_{d}}{2\rho \left(1+\upsilon_{d}\right)}.}$$

Special cares should be needed to evaluate the dynamic mechanical properties of concrete, non-homogeneous but statistically isotropic material, from the elastic wave velocity measurements because different wave velocity measurements ($V_{p\;}$, $V_{c\;}$ or $V_{s\;}$) could result in different values in $E_{d}$ (or $\upsilon_{d}$) estimation [16]. It is difficult to find data in previous researches that investigated the variation of $E_{d}$ (or $\upsilon_{d}$) values of heat-damaged concrete from different wave velocity measurements. 

The main objectives of this study are to investigate, by using the three elastic wave velocity measurements ($V_{p\;}$, $V_{c\;}$ or $V_{s\;}$), the effects of heating and cooling on the static and dynamic residual mechanical properties of 35 MPa (5000 psi) concrete used in the design and construction of nuclear reactor auxiliary buildings in Korea, and to establish the correlation between static and dynamic properties of heat-damaged concrete. In this study, concrete specimens (100 mm × 200 mm cylinder) were fabricated in an actual batch plant at a nuclear power plant construction site in Korea. To induce thermal damages to the concrete, the specimens were heated to target temperatures from 100 °C to 1000 °C with intervals of 100 °C, at a heating rate of 5 °C/min and allowed to reach room temperature by natural cooling. Concrete specimens were heated after at least 91 days of concrete age to ensure the water content of concrete converging into about 3% to 4%. The dynamic properties (i.e., dynamic elastic modulus and dynamic Poisson’s ratio) of the concrete were evaluated by elastic wave measurements (P-wave velocity measurements according to ASTM C597/C597M-16 [17] and fundamental longitudinal and transverse resonance tests according to ASTM C215-14 [18]) before and after the thermal damages. The static properties (compressive strength, static elastic modulus and static Poisson’s ratio) were measured by uniaxial compressive testing in accordance with ASTM C39-14 [19] and ASTM C469-14 [20] after the thermal damages. The significance of this research is to improve understanding of the static and dynamic residual mechanical properties of NPP concrete in Korea under heating and cooling, and to provide fundamental data for developing in-situ NDE methods based on elastic wave measurements for the condition assessment of concrete in structures with heat-induced damages.

## 2. Materials and Sample Preparation

### 2.1. Preparation of Test Samples

The concrete used in this study had a strength of 35 MPa (5000 psi), used for the design and construction of nuclear auxiliary buildings in Korea. The concrete specimens (100 mm by 200 mm cylinder) were fabricated using materials directly sourced from a batch plant located in an actual NPP construction site in Korea. The concrete was made of type 1 Portland cement, siliceous aggregates (natural sand and crushed coarse aggregates in Korea), with water-to-cement ratio of 0.4 and nominal maximum aggregate size of about 25 mm (1 inch). Details of the concrete mix proportion is confidential and not provided in this study. 

Forty-three concrete cylinders were cast in 100 mm by 200 mm plastic molds in accordance with ASTM C31/C31M [21]. Thirty-three cylinders (i.e., three cylinders per each target temperature) were used to evaluate the thermal properties of concrete after heating and cooling with varying target temperature (20 °C for control specimens, and 100 °C to 1000 °C with intervals of 100 °C for test specimens). In addition, ten concrete cylinders were prepared to monitor the internal temperature of concrete during the heating and cooling process. A plastic tube with diameter of 0.5 mm was fitted inside the concrete specimen. The tube served as a casing for placement of the thermocouple inside the specimen. All concrete cylinders were air-cured in a constant temperature-and-humidity room (20 °C, 60% relative humidity) after being de-molded on the day following casting. A series of static and dynamic tests was performed before and after the heating and cooling process. In this study, all concrete specimens were heated after at least 91 days of concrete age, such that the water content of concrete was around 3% to 4%.

### 2.2. Temperature Control

The concrete specimens were heated inside a programmable electric furnace that can control a temperature history for heating (Figure 1a). The electric furnace could simultaneously heat four specimens (Figure 1b). Figure 2a shows the variation of temperatures in the electric furnace, on the surface, and the core of a concrete cylinder heated to the target temperature of 1000 °C. An R-type thermocouple (platinum and platinum-rhodium alloy, 87% Pt and 13% Rh by weight) installed in the electric furnace was used to monitor air temperature: concrete temperature was measured by two K-type thermocouples (chromel and alumel alloy) embedded in concrete. Note that the R-type thermocouple gives stable and accurate (an error of ±1.5 °C) results and is used in high temperature applications (0 °C to 1600 °C) [22], but are relatively expensive; the K-type thermocouple is accurate (an error of ±2.2 °C), used in a wide temperature range (0 °C to 1260 °C) [22] and cost-effective for measuring concrete temperature. The internal temperature of the furnace was controlled to increase linearly to the target temperature, with a constant rate of 5 °C/min. It was observed that the surface temperature of concrete increased at a constant rate, which is compatible with the furnace temperature. However, the core temperature of the concrete increased with several delays, which was at a temperature range of 108 °C to 180 °C and at temperatures about 525 °C and 825 °C. Figure 2b shows the variation of the temperature difference between the surface and the core of the concrete specimen, the temperature history of which is shown in Figure 2a. The timing of the first temperature jump, indicated by ① in Figure 2b, agrees well with that of the first temperature stagnation in Figure 2a. This phenomenon can be explained by the evaporation of water in concrete [5], which also causes a rapid reduction of concrete mass (see Figure 3). In this study, relative mass $\kappa_{m}$ and relative mass density $\kappa_{\rho }$ were determined using Equations (4) and (5), respectively.
(4)$$\kappa_{m}=\frac{m_{T}}{m_{T=20\;{}^{\circ}C}},$$
(5)$$\kappa_{\rho }=\frac{\rho_{T}}{\rho_{T=20\;{}^{\circ}C}},$$
where $m_{T}\;$ and $\rho_{T}$ are mass and mass density of heat-damaged concrete with maximum exposed temperature of T. The other three peaks, indicated by ②, ③ and ④ in Figure 2b, also agree with the temperature stagnation of the core temperature, which appears to be related to chemical changes of cement pastes after exposure to high temperatures [23]. 

In this study, the temperature of the electric furnace was kept constant around the target temperature until the core temperature of concrete caught up with the surface temperature. The heated concrete cylinders were thus presumed to reach the thermally-steady-state condition at target temperatures. Specifically, the steady state condition was defined as the temperature when the core of the concrete was equal to the surface temperature of the concrete, or when the internal temperature increment rate fell below 0.5 °C/min. After reaching the steady state condition, the heat source of the electric furnace was turned off and naturally cooled in an airtight state. In the cooling period, temperature decrease in concrete follows an exponential function [24] (*T* = c_0_e^γt^, where c_0_ is a constant dependent on a target temperature in a unit of °C; γ is cooling rate about −0.14/h for all concrete specimens and the different target temperature levels; *T* is temperature of the concrete and *t* is time in a unit of hour).

## 3. Methods

### 3.1. UPV Test for Constrained Compressive Wave Velocity (P-Wave Velocity) Measurement

The P-wave velocity of concrete, *V_P_*, was measured according to ASTM C597 [17] using a pair of P-wave transducers (see Figure 4a), each of which generates and receives an ultrasonic pulse of about 52 kHz through a concrete cylinder. The source transducer was driven by a 200 V square pulse having a duration of 10 µs using a pulse-receiver (Panametrics 5077 PR). Transient stress waves, which were generated by the source sensor, propagated in the concrete and were measured by the receiving sensor. The received signal was digitized by a high-speed digital oscilloscope (NI-PXI 5101) at a sampling rate of 10 MHz and a total signal length of 0.001 s. The digitized data were transferred to a laptop computer for data storage and post-processing.

Figure 5 shows typical time domain signals measured from concrete cylinders with different degrees of heat-induced damages. The velocity of an ultrasonic wave can be calculated by dividing the wave path *L* over the travel time (*t*) as follows:(6)$$V_{P}\;=\;\frac{L}{t}.$$

To obtain the actual wave travel time *t* through the material, the time delay in the measurement system *t_m_* should be subtracted from the time directly measured from the time signals *t_r_* as follows:(7)$$t\;=\;t_{r}-\;t_{m},$$
where $t_{r}$ is the first arrival time of an ultrasonic pulse (direct P-waves or constrained compressive waves) through a concrete specimen. Note that theoretically, P-waves are faster in time signals than any other refracted and reflected waves from the boundary of concrete cylinders. In this study, an approximate arrival time was first obtained using the conventional threshold method in the literature [25], and an accurate arrival time was calculated by fitting a line to the signal data [26]. The intersection of the fitting line and the calculated zero-signal level defines the P-wave travel time. For the ultrasonic transducers used in this study, time delay *t_m_* (equal to 1.34 µs) caused by coating, cables and equipment was determined by measuring the travel time of an ultrasonic pulse when the tips of the receiver and the source transducers are in contact. In the velocity calculation, the longitudinal length of specimens *L* was measured from concrete cylinders just before the UPV (ultrasonic pulse velocity) measurements.

### 3.2. Free-Free Resonance Frequency Test

The free-free resonance frequency (FFR) test was used to measure unconstrained compressive wave velocity and shear wave velocity of heat-damaged concrete cylinders. Figure 4b illustrates the FFR test set-up for longitudinal and transverse resonance frequency measurements of concrete cylinders in this study [18]. A steel ball having a diameter of 10 mm was used as an impact source for generating incident stress waves in concrete specimens. The steel ball was effective for generating wideband frequency signals from very low to 20 kHz, covering the frequency range of resonance tests in this study. The dynamic response of the concrete cylinder was measured by an accelerometer (PCB 352C33), with ±5% frequency range of 0.5 Hz to 10 kHz and resonance frequency of around 50 kHz [27], attached to the concrete specimen according to ASTM C215-08 [18]. The acquired signals using the accelerometer were stabilized using a signal conditioner (PCB 482C16) and digitized at a sampling frequency of 1 MHz using an NI-USB 6366 oscilloscope. Resulting time signals were converted to the frequency domain using the FFT (fast Fourier transform) algorithm. The resonance frequencies of the cylinders are manifested as dominant amplitudes in the amplitude spectrum. The most dominant frequency was regarded as the fundamental resonance frequency for longitudinal *f_L_* and transverse *f_Tr_* modes.

Figure 6a,b show the variations, with temperature changes, of spectral amplitudes of dynamic response of concrete cylinders obtained from the longitudinal and transverse FFR testing, respectively. It is clearly demonstrated that the first peak frequency shifted to low frequency as the temperature increased from room temperature to 1000 °C. In this study, *V_C_* and *V_S_* were calculated using Equations (8) and (9), respectively.
(8)$$V_{c}=2L\frac{f_{L}}{K_{E}},$$
(9)$$V_{s}=2L\frac{f_{Tr}}{K_{G}},$$
where *f_L_* and *f_Tr_* are fundamental resonance frequencies for longitudinal and transverse modes, respectively, measured by the FFR test; *L* is the length of a concrete cylinder (see Figure 4b); *K_E_* is a correction factor for the dispersion effect of stress waves in a 100 mm × 200 mm concrete specimen (*D*/*L* = 0.5, where D is the diameter of a concrete cylinder) as follows [28]:(10)$$K_{E}=-0.095\upsilon_{d}+1.01252,$$
*K_G_* is a correction factor converting the transverse resonance frequency to the torsional resonance frequency for a 100 mm × 200 mm concrete specimen (*D*/*L* = 0.5), as follows [28]:(11)$$K_{G}=0.337\upsilon_{d}+0.8833,$$
note $\upsilon_{d}$ in Equations (10) and (11) is dynamic Poisson’s ratio of concrete.

### 3.3. Uniaxial Compression Test

After completing the nondestructive testing at each target temperature, static residual mechanical properties (compressive strength, static elastic modulus and static Poisson’s ratio) of concrete cylinders were measured using a universal testing machine (UTM) with a capacity of 2000 kN according to ASTM C39/C39M-14 [19] and ASTMC469/C469M-14 [20], respectively (Figure 7). Tests were performed at a stress rate of approximately 0.28 MPa/s. Longitudinal deformations were measured by using two sets of extensometers attached to two fixed frames (① Figure 7). It had two aluminum rings with screws for attachment to the specimen (③ and ④ in Figure 7). The spacing with screws on the top and bottom rings was 100 mm, which serves as a gauge length to calculate axial strain from the measured deformations. The transverse deformation was measured by using an extensometer (② in Figure 7) attached to one end of a fixed frame with a rotating pivot at the other end. The distance from the pivot to the center of the cylinder is one-third of the distance to the transverse extensometer. Therefore, radial deformation due to specimen deformation was calculated by multiplying one-third and measured deformation from the transverse extensometer. The load and deformation data were measured by a data logger (DEWE43A) with a sampling frequency of 100 Hz.

The static elastic modulus of concrete is defined as a chord modulus from the stress-strain curve, calculated as follows [20],
(12)$$E_{s}=\frac{0.4F_{c}-\sigma \left(\varepsilon_{1}\right)}{\varepsilon_{L2}-\varepsilon_{L1}},$$
where $\varepsilon_{L2}$ is the longitudinal strain corresponding to the 40% of maximum stress (*F_c_*) and $\varepsilon_{L1}$ is the longitudinal strain of 0.00005. In this research, the first and second points were determined by a linear regression of the local data in the measured stress-strain curves. 

The static Poisson’s ratio of concrete $\upsilon_{s}$ is defined as the slope of two points in the transverse and longitudinal strain curves, with a first point at a longitudinal strain level of 0.00005 (*ε*_1_) and the second point corresponding to 40% of maximum stress (*F**_c_*), calculated as follows [20]:(13)$$\upsilon_{s}=\frac{\varepsilon_{T2}-\varepsilon_{T1}}{\varepsilon_{L2}-\varepsilon_{L1}},$$
where $\varepsilon_{T2}$ and $\varepsilon_{T1}$ are the transverse strains at mid-height of the specimen corresponding to 40% of maximum stress (*F**_c_*) and longitudinal strain of 0.00005, respectively. 

## 4. Results and Discussion

### 4.1. Stress Wave Velocities in Heat-Damaged Concrete

Figure 8 shows that the square of the three different velocities (constrained compressive wave velocity *V_P_*, unconstrained compressive wave velocity *V_C_*, and shear wave velocity *V_S_*) in heat-damaged concrete decreases as temperature increases, with a bilinear relationship. Note that *V_C_* and *V_S_* were determined using Equations (7) and (8), respectively, based on a Poisson’s ratio of 0.2. The velocity reduction curve can be divided into two sections according to the reduction rate with respect to temperature. As the temperature increased from 20 °C to 600 °C, the square values of *V_P_*, *V_C_*, and *V_S_* almost linearly decreased to about 15% (2.18 [km/s]^2^/14.10 [km/s]^2^), 10% (1.35 [km/s]^2^/12.57 [km/s]^2^) and 12% (0.64 [km/s]^2^/5.25 [km/s]^2^) of the initial values at room temperature, respectively. Further increase of the temperature from 600 °C to 1000 °C decreased the square values of the three velocities, with a slower reduction rate, and dropped the values to almost zero at 1000 °C.

Abrupt decrease of elastic wave velocities of concrete (*V_P_*, *V_C_*, and *V_S_*) shown in Figure 8 can be explained by damage of concrete in the microstructure level, i.e., decomposition of cement paste and expansion of aggregates in concrete. After the temperature of concrete increases between 400 °C to 550 °C, the degradation of calcium hydroxide (Ca(OH)_2_ → CaO + H_2_O) causes shrinkage of concrete and changes in the microstructure of cement pastes. At the temperature between 500 °C to 600 °C, the phase of quartz (SiO_2_) in aggregates converts from α to β, which causes a volume expansion of about 0.45%, and promotes microcracking with increasing internal stress of concrete, especially for siliceous aggregate concrete [23]. As will be discussed in Section 4.3 and Section 4.4, these microstructural changes in concrete affect static and dynamic elastic moduli of heat-damaged concrete in a similar way.

### 4.2. Poisson’s Ratio

Figure 9 shows the variation, with increasing temperature, of static and dynamic Poisson’s ratios of concrete exposed to high temperature and cooling to room temperature. In this study, dynamic Poisson’s ratio of concrete was evaluated based on the three elastic wave velocities. Rearranging Equations (1)–(3) results in three dynamic Poisson’s ratios as follows: (14)$$\upsilon_{d,cs}=\frac{\gamma_{1}{}^{2}}{2}\;-1,$$
(15)$$\upsilon_{d,PS}=\frac{\gamma_{2}{}^{2}-2}{2\gamma_{2}{}^{2}-2}\;,$$
(16)$$\upsilon_{d,PC}=\frac{1-\gamma_{3}{}^{2}+\sqrt{{\left(\gamma_{3}{}^{2}-1\right)}^{2}+8\gamma_{3}{}^{2}\left(\gamma_{3}{}^{2}-1\right)}}{4\gamma_{3}{}^{2}},$$
where $\upsilon_{d,CS}$, $\upsilon_{d,PS}$ and $\upsilon_{d,PC}$ are the three dynamic Poisson’s ratios measured from *V_C_* and *V_S_*, *V_P_* and *V_S_*, and *V_P_* and *V_C_*, respectively; γ_1_, γ_2_, and γ_3_ are *V_C_*/*V_S_*, *V_P_*/*V_S_* and *V_P_*/*V_C_*, respectively. For comparison, static Poisson’s ratio $\upsilon_{s}$ determined using Equation (13) based on the uniaxial compression test is shown in the same figure. At room temperature, the static and dynamic Poisson’s ratios were about 0.15 and 0.18, respectively, which are normal values for normal strength concrete [29]. Even when the concrete cylinders were exposed to higher temperatures in the range of 20 °C to 400 °C, the static and dynamic Poisson’s ratios in this study did not change much, compared to the initial Poisson’s ratio at room temperature. However, in the temperature range of 500 °C to 600 °C, the static and dynamic Poisson’s ratios decreased to about one-third of the initial values at room temperature. The gradual decrease in Poisson’s ratios can be explained by the development of micro-cracks in the cement paste and interfacial transition zone of aggregates as the temperature increased [30]. However, in a temperature range from 700 °C to 1000 °C, the static and dynamic Poisson’s ratios show high variability with a coefficient of variation over 1.0; some results were unreasonable (greater than 0.5 or negative value). Therefore, it was difficult to reach general conclusions of the variation of Poisson’s ratio above temperatures of 700 °C in this study.

### 4.3. Dynamic Elastic Modulus

Figure 10 shows the variation of dynamic elastic modulus of concrete with increasing temperature. In this study, dynamic elastic modulus was determined from the fundamental longitudinal resonance frequency ($f_{L}$) using an equation described in ASTM C215 [18] as follows,
(17)$$E_{d,LR}=\beta_{L}Mf_{L}^{2}\;(\mathrm{Pa}),$$
where *β_L_* is a constant dependent on dimensions and Poisson’s ratio of concrete specimens ($\mathrm{equal}\;\mathrm{to}\;5.093\frac{L}{D^{2}}$) for a cylinder in N·s^2^ (kg·m^2^), M is the mass of specimens in kg and D is the diameter of a cylinder in m. The dynamic elastic moduli gradually decreased with temperature, and at 1000 °C, dropped by about 90% of sound concrete. $E_{d,LR}$ values linearly decrease as the temperature increased from 20 °C to 600 °C. At 600 °C, $E_{d,LR}$ values are only about 7% of the initial values at room temperature. After 600 °C, $E_{d,LR}$ values further decreased with temperature, with a slower reduction rate, and dropped to almost zero at 1000 °C.

For comparison, three dynamic elastic moduli were evaluated from the three elastic wave velocities using Equations (18)–(20).
(18)$$E_{d,P}=\frac{\rho \left(1+\upsilon_{d}\right)\left(1-2\upsilon_{d}\right)}{\left(1-\upsilon_{d}\right)}V_{P}^{2},$$
(19)$$E_{d,C}=\rho V_{C}^{2},$$
(20)$$E_{d,S}=2\rho \left(1+\upsilon_{d}\right)V_{S}^{2},$$
where *E_d,P_*, *E_d,C_* and *E_d,S_* are dynamic elastic moduli of concrete based on *V_p_*, *V_c_* and *V_s_*, respectively; and $\upsilon_{d}$ is dynamic Poisson’s ratio of concrete. In Figure 10, the three velocity dynamic elastic moduli (*E_d,P_*, *E_d,C_*, and *E_d,S_*) are shown as open circles with black, blue, and red colors, respectively. Note that *E_d,P_* and *E_d,S_* values in Figure 10 were determined using Poisson’s ratio of 0.2.

It is of interest to investigate the effect of Poisson’s ratio on the variation of the three dynamic moduli based on elastic wave velocities with temperature. Table 1 summarizes RMSE (root mean square error) between the three velocity dynamic elastic moduli and the standard dynamic elastic modulus (*E_d,LR_*) with different Poisson’s ratios. First, *E_d,C_* values are close to *E_d,LR_* values, with RMSE (root mean square error) of 0.31 GPa, regardless of Poisson’s ratio. In contrast, *E_d,P_* and *E_d,S_* values based on Poisson’s ratio of 0.2 slightly deviated from *E_d,LR_* values, with RMSE of 1.04 GPa and 0.63 GPa, respectively. It was observed that the *E_d,P_* and *E_d,S_* values determined on the basis of measured dynamic Poisson’s ratios (*υ**_d_**,_CS_*, *υ**_d_**,_PS_* or *υ**_d_**,_PC_*) resulted in smaller RMSE (see Table 1). Incidentally, it was observed that dynamic elastic modulus rapidly decreased as temperature increased from 20 °C to 600 °C, with sensitivity of about 0.041 GPa/°C. Consequently, approximate ranges of errors in temperature estimation based on *E_d,P_* and *E_d,s_* based on the constant Poisson’s ratio of 0.2 would be within ±25 °C and ±15 °C, respectively. Therefore, it is reasonable to say that Poisson’s ratio does not greatly affect the variation of dynamic moduli with temperature.

### 4.4. Static Elastic Modulus

Figure 11 shows the variation of static elastic modulus reduction coefficient $\kappa_{E}$ of concrete with increasing temperatures. Consistent with observations of previous researches [31,32], the reduction of static elastic modulus of concrete with temperature followed a bilinear relationship. Figure 11 also compares the $\kappa_{E}$ values measured in this study and the reduction models incorporated in the famous concrete fire design standards [33,34,35]. The reduction curve in this study was comparable to the model curve from ACI 216 [26], which is based on ‘test cold’. In contrast, the values of the chord-elastic modulus based on ASTMC469/C469M-14 [20] was determined from the temperature-dependent constitutive models described in the codes [34,35]. For the same temperature, the experimental results in this study were about 20% to 30% higher than the tested “hot” properties predicted by EN 1992-1-2 [35] and ASCE [34]. This result appears to be attributed to the stiffness recovery of concrete after cooling to room temperature, which is regarded as the positive effect of cooling.

### 4.5. Relationship between Static and Dynamic Elastic Moduli

Figure 12 compares the variation of static elastic modulus of concrete with dynamic elastic modulus. Overall, the static elastic modulus appeared to be slightly smaller than dynamic elastic modulus in the temperature range of 20 °C to 1000 °C. At room temperature, the static modulus of concrete was about 20% smaller than that of the dynamic elastic modulus. The dynamic elastic modulus gradually drew closer to the static elastic modulus as the elastic modulus decreased (or as temperature increased), and both moduli converged to zero. Based on these experimental features, the expression relating the static to dynamic elastic moduli of heat-damaged concrete was obtained by non-linear regression using the following power function relationship:(21)$$E_{d}\left(T\right)=\alpha E_{s}{\left(T\right)}^{b},$$
where $E_{s}\left(T\right)$ and $E_{d}\left(T\right)$ are the static and dynamic elastic moduli of concrete exposed to the maximum temperature of T in the heating and cooling process, in GPa; and *α* and *b* are best-fit constants according to regression analysis (see Figure 12), which was performed using the Nonlinear Least Squares method and Trust-Region algorithm in the MATLAB curve fitting toolbox. In this study, the best-fit curve was obtained at *α* = 1.887 and *b* = 0.8455, with the least R-square of 0.98. The RMSE between the measured and predicted static elastic modulus using the proposed curve was 0.55 GPa, which shows good agreement between the experiments and the model’s predictions. 

Figure 12 compares the dynamic and static elastic moduli in this study with empirical models proposed by previous researchers [36,37] at room temperature. Lydon and Balendran proposed an empirical relationship (*E**_s_* = 0.83*E_d_*) and Popovics [37] proposed a more general relationship for both lightweight and normal density concrete, taking into account the effect of concrete density ($446.09\frac{E_{d}^{1.4}}{\rho }$ in GPa, where ρ is the density of the hardened concrete in kg/m^3^). Interestingly, although proposed at room temperature, the static and dynamic moduli relations proposed by Lydon and Balendran [36] and Popovics [37] showed good agreement with the experimental results of heat-damaged concrete in this study. Particularly, the equation proposed by Lydon and Balendran [36] showed a fairly good agreement with the experimental data in this study, with RMSE of 0.68 GPa, comparable to that of the best-fit curve in this study. Therefore, it could be seen that the Lydon-and-Balendran equation and the best-fit curve is effective for representing the relationship between static and dynamic elastic moduli of heat-damaged concrete in this study.

### 4.6. Compressive Strength

Figure 13 shows that the compressive strength reduction coefficient of concrete, $\kappa_{Fc}$, appears to decrease linearly as the maximum exposed temperature *T* increased from 20 °C to 1000 °C. At 1000 °C, the concrete lost about 95% of initial strength at 20 °C. Figure 13 also compares the $\kappa_{Fc}$ values measured in this study and the strength reduction curves incorporated in the famous concrete fire design standards [33,34,35]. The $\kappa_{Fc}$ values measured in this study were between the residual strength curves for siliceous and calcareous aggregate concrete after heating and cooling to room temperature (tested ‘cold’), as described in ACI 216 [33]. In contrast, strength reduction curves incorporated in EN 1992-1-2 [35] and ASCE [34], which were established based on experimental data at hot conditions (tested ‘hot’), tended to overestimate the residual strength of concrete. The additional strength reduction of concrete after cooling (also called ‘cooling effect’) was attributed to different volume changes in cement paste, and aggregates and thermal gradients in concrete during cooling.

Figure 14 shows the correlation between the static elastic modulus and compressive strength of the heat-damaged concrete. Figure 14 also compares the experimental results with code equations [38,39,40] and a practical equation proposed by Noguchi et al. [41] established at room temperature. The EN 1992-1-1 [40] presented a correlation between the static elastic modulus *E_s_* and compressive strength *F_c_* for both normal-strength and high-strength concrete:(22)$$E_{s,EN1992-1-1}=22,000\;{\left(F_{c}/10\right)}^{\frac{1}{3}}\left(\mathrm{MPa}\right).$$

ACI 318 [39] and ACI 363 [38] committees proposed two equations according to the compressive strength of concrete: (23)$$E_{s,ACI318}=0.043\;F_{c}{}^{1/2}\sqrt{F_{c}}\rho_{c}{}^{3/2}\left(\mathrm{MPa}\right)\;(F_{c}\leq \;38\;\mathrm{MPa}),$$
(24)$$E_{s,ACI363}=\left(3320\;F_{c}{}^{\frac{1}{2}}+6900\right){\left(\frac{\rho_{c}}{2300}\right)}^{\frac{3}{2}}\left(\mathrm{MPa}\right)\;(21\;\mathrm{MPa}\leq F_{c}\leq \;83\;\mathrm{MPa}).$$

Noguchi et al. [41] proposes a practical equation of elastic modulus of concrete based on the results of normal and high strength concrete: (25)$$E_{s,Noguchi}=k_{1}k_{2}33,500{\left(\frac{F_{c}}{60}\right)}^{\frac{1}{3}}{\left(\frac{\rho_{c}}{2300}\right)}^{2}\;\left(\mathrm{MPa}\right),$$
where *k*_1_ and *k*_2_ are a correction factor for the type of coarse aggregates and supplementary cementitious materials, respectively. The static elastic modulus predicted by the equations, proposed by ACI 363 committee [38] and Noguchi et al. [41], was in fairly good agreement with the experimental results at room temperature, in this study. However, for the same compressive strength, the residual static elastic modulus of heat-damaged concrete was lower with increasing temperature, compared to the value estimated by the equations at room temperature. This phenomenon indicates that the elastic modulus of heat-damaged NPP concrete in Korea decreases more rapidly than the compressive strength of concrete with increasing temperature. According to a previous researcher [42], this appears to be mainly due to the destruction of bonds caused by the expansion of siliceous aggregates in Korea NPP concrete with increasing temperature. 

An approximate expression relating the static elastic modulus and compressive strength was obtained by non-linear regression using the following power function relationship:(26)$$E_{S}\left(T\right)=aF{\left(T\right)}^{b}.$$

Regression analysis was performed using the Nonlinear Least Squares method and the Trust-Region algorithm in the MATLAB curve fitting toolbox. In this study, the best-fit curve was obtained at *a* = 0.01425 and *b* = 1.994, with the least R-square of 0.9826. The RMSE between the measured and predicted static elastic modulus using the proposed curve was 1.148 GPa, which shows good agreement between the experiments and the model’s predictions. 

## 5. Summary and Conclusions

A series of experimental studies was performed to investigate the effects of heating and cooling on the static and dynamic residual properties of 35 MPa (5000 psi) concrete used in the design and construction of nuclear reactor auxiliary buildings in nuclear power plants (NPPs) in Korea. The specific conclusions obtained from this study are summarized as follows:
(1)Residual static mechanical properties (i.e., compressive strength and elastic modulus) of NPP concrete in Korea decreases as maximum exposed temperature increases. However, it was observed that exposure to high temperature affects the two residual mechanical properties of concrete in different ways: The heat-induced damages decrease the elastic modulus of concrete more rapidly than the compressive strength of concrete. The residual compressive strength decreases linearly with increasing temperature from 20 °C to 1000 °C. At 1000 °C, the residual compressive strength is about 10% of the initial strength at room temperature. In contrast, the residual static elastic modulus decreases more rapidly with increasing temperature from 20 °C to 600 °C. At 600 °C, the residual modulus is only 10% of the initial value.(2)The residual dynamic elastic modulus of concrete decreases as the severity of heat-induced damage increases. At the same temperature, the dynamic elastic modulus was slightly greater than the static elastic modulus. The relationship between static and dynamic elastic moduli of heat-damaged concrete was established in this study. An interesting finding in this study is that the experimental results of heat-damaged concrete are in good agreement with famous equations relating static and dynamic elastic moduli of solid concrete at room temperature.(3)The three wave velocities (unconstrained and constrained compressive wave velocities and shear wave velocity) of heat-damaged concrete are mainly dominated by residual dynamic elastic modulus: the square of the wave velocity of concrete is proportional to the dynamic elastic modulus of the concrete. This relationship is consistent with the elastic wave propagation theory established for isotropic and homogenous solid materials.(4)This study is based on the results of experiments using concrete employed at an actual nuclear power plant construction site in Korea. Therefore, this study could contribute to better understanding of the static and dynamic mechanical properties of heat-damaged concrete in NPPs in Korea. However, this study is based on experiments in a laboratory using a single concrete mixing proportion (35 MPa NPP concrete in Korea) and only one type of heating and cooling process. Therefore, more studies based on various concrete mixture proportions and heating and cooling scenarios are needed to gain generality of the results in this study.

## Figures and Tables

**Figure 1 materials-12-02695-f001:**
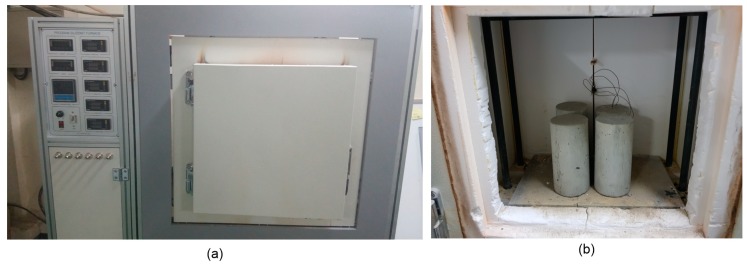
The electric furnace used for heating and cooling of concrete cylinders in this study: (**a**) a control unit and exterior view of the furnace, and (**b**) inside the furnace.

**Figure 2 materials-12-02695-f002:**
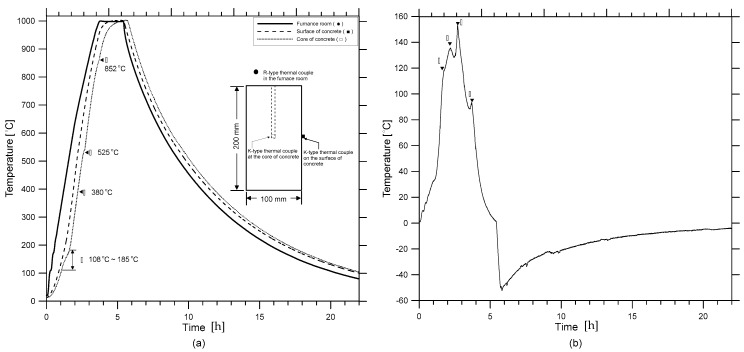
Temperature development and thermal gradient inside concrete exposed to maximum temperature of 1000 °C: (**a**) temperature development, and (**b**) thermal gradient between the surface and core of a concrete cylinder.

**Figure 3 materials-12-02695-f003:**
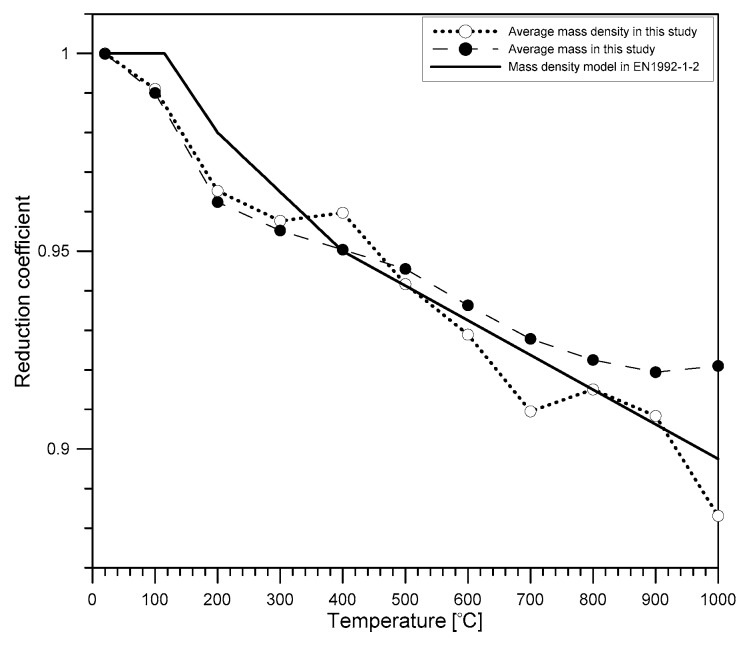
Variation of the relative mass and mass density of concrete with increasing temperature.

**Figure 4 materials-12-02695-f004:**
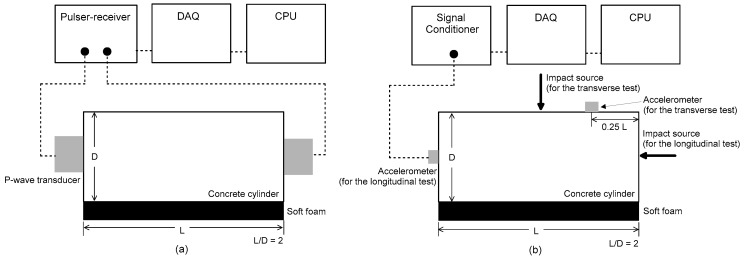
Test setup of evaluating dynamic properties of concrete cylinders for: (**a**) the ultrasonic pulse velocity test and (**b**) the resonance vibration frequency test.

**Figure 5 materials-12-02695-f005:**
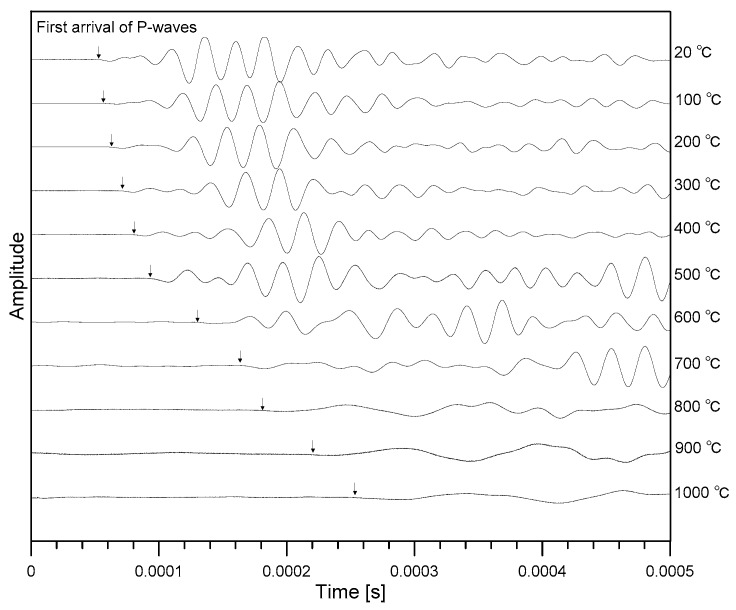
Typical time signals of ultrasonic pulse waves propagating through heat-damaged concrete after exposure to various target temperatures.

**Figure 6 materials-12-02695-f006:**
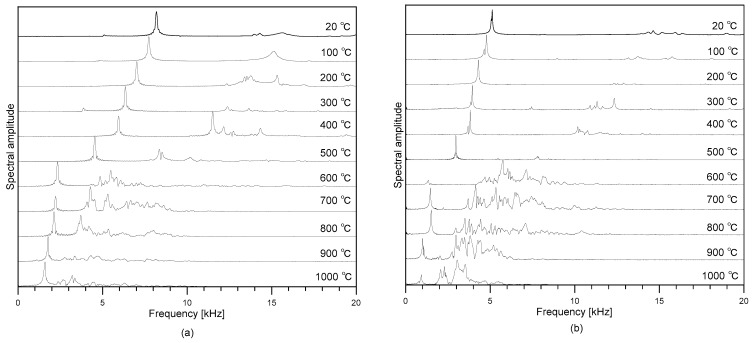
The variations of spectral amplitudes of dynamic response of concrete cylinders obtained from: (**a**) longitudinal and (**b**) transverse resonance vibration testing.

**Figure 7 materials-12-02695-f007:**
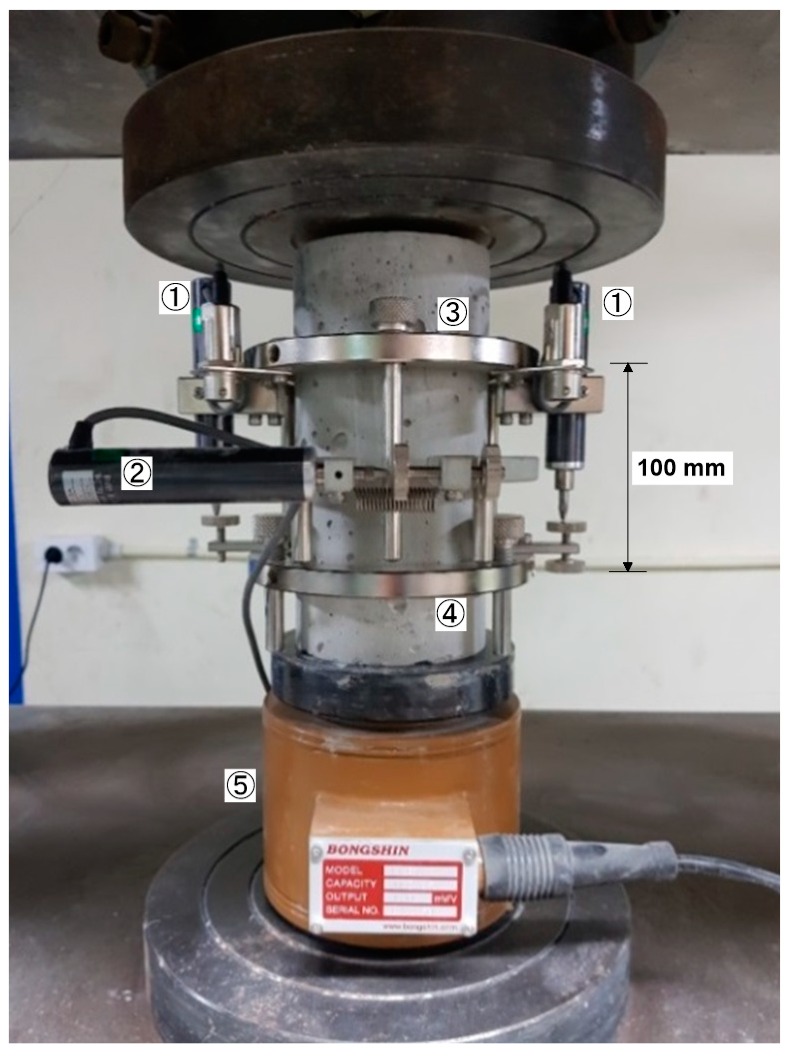
Test setup for the uniaxial compressive test for measurements of compressive strength, static elastic modulus and static Poisson’s ratio of concrete cylinders: ① extensometers for measuring longitudinal deformation; ② an extensometer for measuring transverse deformation; ③ and ④ upper and lower fixed aluminum ring for installing extensometers, respectively; and ⑤ load cell.

**Figure 8 materials-12-02695-f008:**
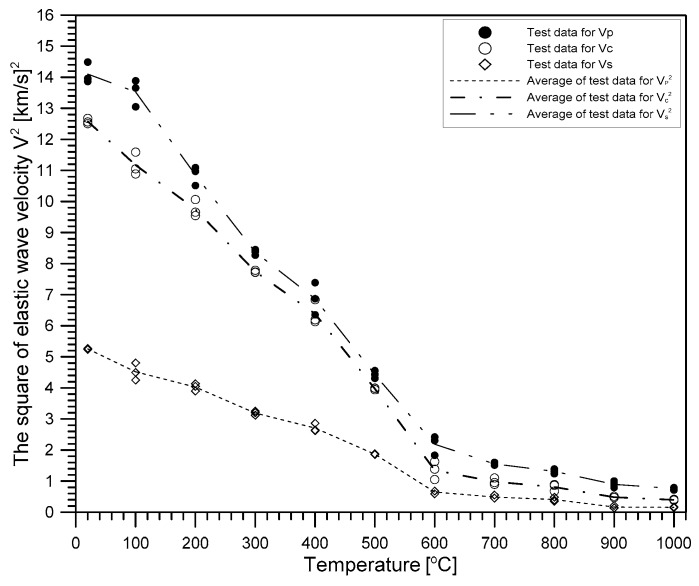
Variation of the square values of the three stress wave velocities (constrained compressive wave velocity, *V_P_*, unconstrained compressive wave velocity, *V_C_* and shear wave velocity, *V_S_*) of heat damaged concrete with maximum exposed temperature.

**Figure 9 materials-12-02695-f009:**
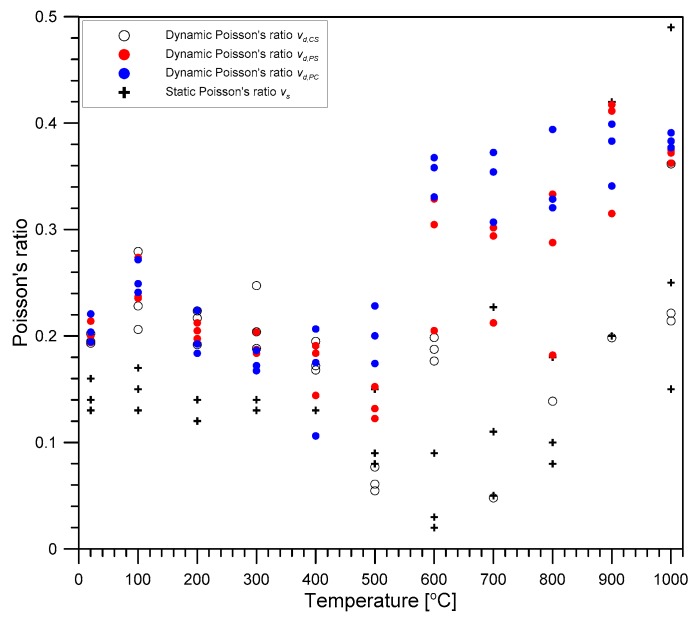
The variation of static and dynamic Poisson’s ratios of concrete with temperature.

**Figure 10 materials-12-02695-f010:**
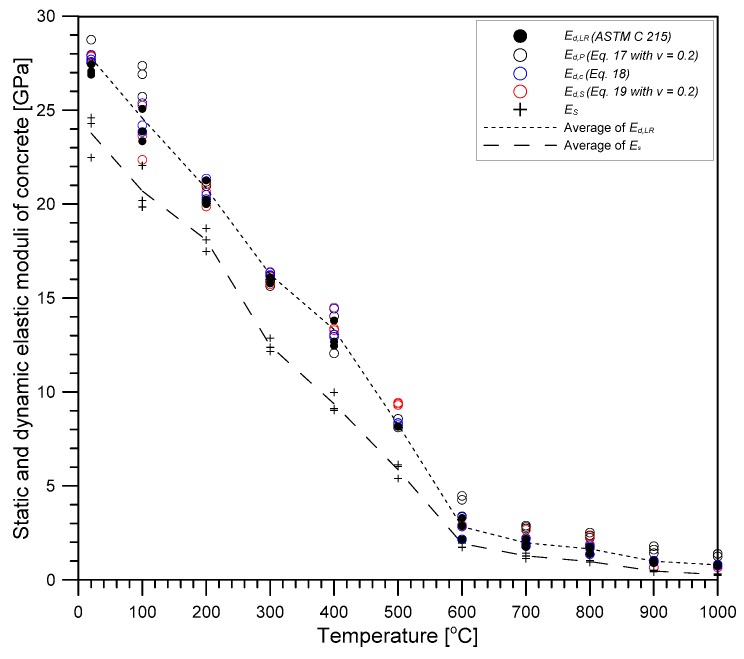
The variation of dynamic- and static elastic moduli of concrete with temperature. The dynamic elastic moduli of *E_d,P_* and *E_d,S_* was based on the constant Poisson’s ratio of 0.2.

**Figure 11 materials-12-02695-f011:**
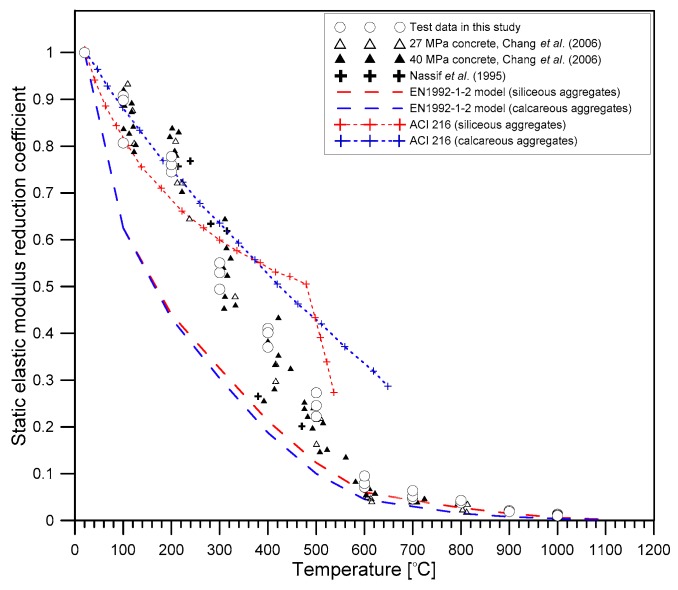
The variation of reduction coefficient in the static elastic modulus of concrete with temperature.

**Figure 12 materials-12-02695-f012:**
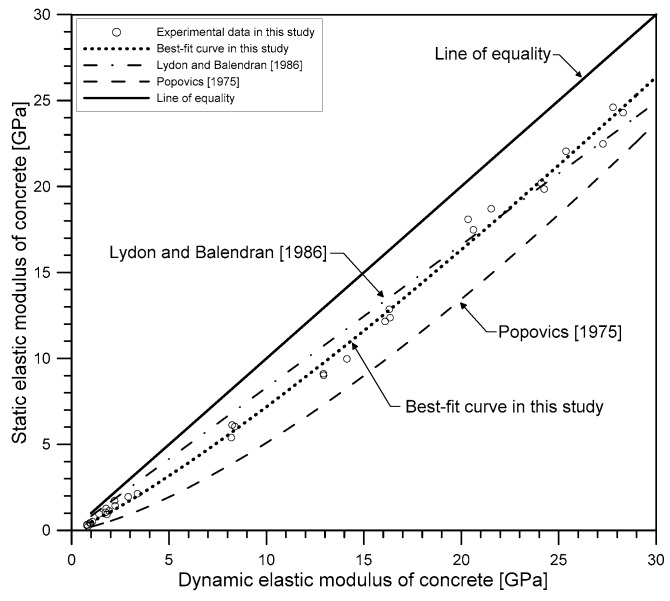
The relationship between static and dynamic elastic moduli of heat-damaged concrete.

**Figure 13 materials-12-02695-f013:**
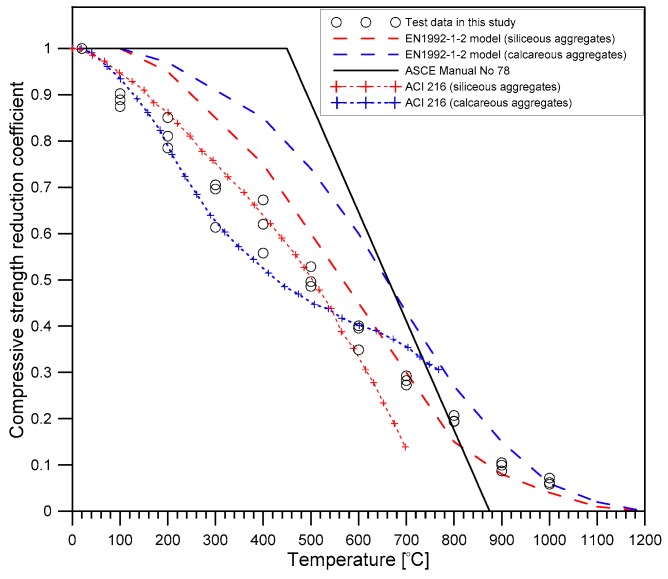
Variation of the reduction coefficient in the compressive strength of concrete with temperature.

**Figure 14 materials-12-02695-f014:**
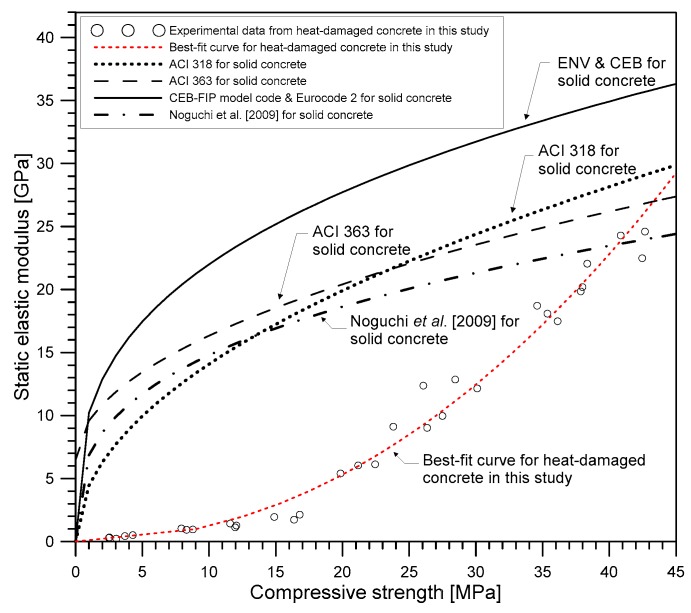
Relationship between the static elastic modulus and the compressive strength of heat-damaged concrete.

**Table 1 materials-12-02695-t001:** RMSE values between the three velocity dynamic elastic moduli (*E_d,C_*, *E_d,P_* or *E_d,S_*) and the standard dynamic elastic modulus (*E_d,LR_*) with different Poisson’s ratios.

Dynamic Elastic Moduli	RMSE Compared to *E_d,LR_* [GPa]			
	*υ* = 0.2	*υ_d_,_CS_*	*υ_d_,_PS_*	*υ_d_,_PC_*
*E_d,C_*	0.31	0.31	0.31	0.31
*E_d,P_*	1.049	0.988	0.543	0.313
*E_d,S_*	0.628	0.392	0.543	0.731

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
