# Peer review of "Evaluation of Static and Dynamic Residual Mechanical Properties of Heat-Damaged Concrete for Nuclear Reactor Auxiliary Buildings in Korea Using Elastic Wave Velocity Measurements"

_materials, 2019, doi:10.3390/ma12172695_

Round 1
Reviewer 1 Report
The manuscript entitled "Evaluation of static and dynamic residual mechanical properties of heat-damaged concrete for nuclear reactor auxiliary buildings in Korea using Elastic wave velocity measurements” investigated the effects of heating and cooling on the static and dynamic residual properties of concrete with a compressive strength of 35 MPa used in design and construction of nuclear reactor auxiliary buildings in Korea. Furthermore, a correlation between the static and dynamic properties of heat-damaged concrete was established. The manuscript contains a clear and valuable contribution to the state of knowledge. However, the manuscript could benefit greatly from professional editing to improve technical writing and English.
After a careful review, the manuscript is good in quality and therefore this reviewer recommends accepting it after considering the following comments.
Technical comments:
1- The authors should increase their discussion on previous related research and highlight how their study is providing a different approach or adding significantly to what has been done.
2- Lines 29-31: What was the cooling rate to reach the room temperature?
3- Lines 32 and 122: “the concrete” should be “concrete” without “the”.
4- Lines 37-38: highlights and important findings should be emphasized in the abstract.
5- Line 83: the symbol does not match with the symbol in equations 1 and 2.
6- Line 137: Is there a reason to wait for that time (91 days)?
7- Line 143: more details about the two types should be provided. Also, what is the difference between the two types?
8- Line 153: more details and explanation should be provided in Figure 3. How did you calculate experimentally the relative mass and mass density?
9- Line 168: the reference of this function should be cited.
10- Figure 2(b): What is the main purpose of the double-sided arrow on this figure?
11- Line 188: “total” should be "a total".
12- Line 192: “travel time” should be written as “travel time (t)”.
13- Lines 200-201: what you mean by “the UPV 200 measurements”? Could you show this length in Figure 4?
14- Line 218: What you mean by very low? Is it 1 or 0.1 kHz?
15- Symbols in equations 7-10 should be clearly defined.
16- Line 250: is it "loading rate" or "stress rate"?
17- Figure 7: some details on the figure can be added to make it clearer.
18- Lines 271-272: What is the difference between transverse and lateral strains? Are both represent one curve or two curves?
19- Figure 9: the legend in this figure does not match with the data presented in the figure. Why?
20- Line 324: what is the meaning of "D"?
21- Line 326: Ed,LR does not match with the symbol in Equation 16. Why?
22- Line 358: I think you mean Figure 11 not Figure 12?
23- Equations 20 and 24: what do you mean by (ɵ) in this equation?
24- Line 381: what do you mean by “chord elastic modulus”? Do you mean “static elastic modulus”?
25- Line 383: I think you mean Figure 12 not Figure 11?
26- Line 403: The sentence “the residual compressive strength” does not match with the title of the vertical axis in Figure 13. Do you mean “compressive strength reduction coefficient”?
27- Line 406: I think you mean Figure 13.
28- Line 439: “compressive concrete” should be “compressive strength of concrete”.
Author Response
The authors deeply appreciate a careful review of the reviewer, and completely agree with all the comments of the reviewer. The manuscript was revised according to the reviewer’s comments. Responses to individual reviewer's comments are provided in a separated file. In addition, a major revision was reflected in the manuscript.

Reviewer 2 Report
This paper reports on the experimental analysis of concrete under different temperature exposure.
The topic of the paper is interesting. However, the remarks below are provided in order to improve the paper.
(Both editorial and technical comments are given in the order they appear in the text.)
- pag 2, line 46-> Please define the acronym NPP. Is it Nuclear Power Plant?
- pag 3, line 122-> I would suggest to give more information about concrete characteristics: mix-design, type of aggregates, water/cement ratio etc.
- pag 5, Figure 2-> It would be interesting to have a discussion on how the temperature time-history modify the concrete characteristics. Indeed, reaching 800°C with a linear monotonic behavior or reaching it with a non-monotonic trend have a different effect on material. Did the authors tried different temperature time-history? What is their opinion on this issue?
- pag 6, line 200, equation 5-> The velocity is measured with equation 5 consider a direct trajectory of the waves. But how can the authors exclude from the signal the refraction and reflection of the waves? Please clarify this point.
- pag 6, line 221-> Please change (16) into [16].
- pag 8, equation (11)-> Please change “fc” into “Fc”. Then, please define the meaning of epsilon_2 and epsilon_1 near equation (11).
- pag 8, line 271-> Please introduce in the text the poisson coefficient symbol \nu.
- pag 10, Figure 9-> Why does the average of dynamic poisson’s ratio V_(d,CS) curve have a peak corresponding to 600 °C? It is a quite different behavior in comparison to the other curves. Please give a sound justification for this result.
- pag 10 , equations (17-19)-> Is \nu \representing the static poisson’s ratio? Please clarify.
- pag 11, table 1-> Why is the E_d,p row presenting the highest RMSE in comparison with the others? Please clarify this point from the mechanical point of view.
- pag 12, Figure 11-> What is the “Model proposed in this study”? Where is the equation of this bilinear line? Please clarify.
- pag 12, line 382, 383-> What is the adopted algorithm for the nonlinear regression? Please give all the details.
- pag 13, line 407-408-> What is the residual strength curve in this study? Where is the equation and the curve? Figure 13 presents just the experimental results. Please clarify.
- pag 14, line 443-> What is the adopted algorithm for the nonlinear regression? Please give all the details.
- pag 15, Figure 14-> Please change the label to denote the curve proposed by the authors. In the current version its name is “line/Scatter plot 22”.
- pag 15, line 479-482-> “The equation is also extended to estimate the relationship between constrained compressive wave velocities (P-wave velocity) of heat-damaged concrete and dynamic elastic modulus of concrete.” Where is this extension? In what part of the manuscript did the authors develop this analysis? Please clarify.
- Finally a general drawback of this study is that the authors considered just one kind of concrete and just one time-history of temperature. In my opinion, this strongly limit the generality of the obtained results.
Author Response

(The authors gave the same response as above.)

Reviewer 3 Report
Row 107 – Few additional information would be beneficial here. When were the specimens heated (meaning at what age) and what was their humidity (please add information)?
Row 108 - What was the cooling method? Was it rapid cooling or gradual cooling?
In the mix design part, please at least refer to the type of aggregate used for mixes, since its properties could affect greatly the test results.
Please explain what happens on the microstructure level (chemical changes) at 600 °C and what do you think causes the abrupt changes of trends shown in figures 8, 10, 11.
Figure 14 – why do you think was the reason your data formed a concave curve while ACI, CEB-FIP and other models clearly show convex curve behaviour? Can you please give some possible explanations? Were these models (ACI, CEB-FIP, etc.) developed for heat-damaged concrete? If so please explain this significant difference, if not please explain why you have compared them to heat-damaged concrete.
Row 457 – you say here that residual strength at 1000°C is 35-45% of initial strength, while figure 13 shows its around 10%
Author Response

(The authors gave the same response as above.)

Round 2
Reviewer 2 Report
This paper reports on the experimental analysis of concrete under different temperature exposure.
The topic of the paper is interesting and my questions were clearly answered.
The paper has been strongly improved and can be accepted in this form.